# SARS-CoV-2 Delta AY.1 Variant Cluster in an Accommodation Facility for COVID-19: Cluster Report

**DOI:** 10.3390/ijerph19159270

**Published:** 2022-07-28

**Authors:** Takayuki Ohishi, Takuya Yamagishi, Hitomi Kurosu, Hideaki Kato, Yoko Takayama, Hideaki Anan, Hiroyuki Kunishima

**Affiliations:** 1Department of Infection Control and Prevention, Saiseikai Yokohama Eastern Tobu Hospital, 3-6-1 Shimosueyoshi, Tsurumi-ku, Yokohama 230-8765, Japan; 2Kanagawa Prefectural Government, 1, Nihonodori, Naka-ku, Yokohama 231-0021, Japan; ekato@mac.com (H.K.); yoko@med.kitasato-u.ac.jp (Y.T.); anan@za3.so-net.ne.jp (H.A.); h2kuni@marianna-u.ac.jp (H.K.); 3Antimicrobial Resistance Research Center, National Institute of Infectious Diseases, 1-23-1 Toyama, Shinjuku Ward, Tokyo 162-8640, Japan; tack-8@nih.go.jp (T.Y.); hitomik@nih.go.jp (H.K.); 4Infection Prevention and Control Department, Yokohama City University Hospital, 3-9 Fukuura, Kanazawa-ku, Yokohama 236-0004, Japan; 5Department of Infection Control and Infectious Diseases, Research and Development Center for New Medical Frontiers, Kitasato University School of Medicine, 1-15-1 Kitazato, Minami-ku, Sagamihara 252-0374, Japan; 6Department of Infection Control and Prevention, Kitasato University Hospital, 1-15-1 Kitazato, Minami-ku, Sagamihara 252-0375, Japan; 7Fujisawa City Hospital, 2-6-1 Fujisawa, Fujisawa 251-8550, Japan; 8Infectious Diseases Course, St. Marianna University School of Medicine, 2-16-1 Sugao, Miyamae-ku, Kawasaki 216-8511, Japan

**Keywords:** SARS-CoV-2, Delta AY.1, variant cluster

## Abstract

Background: This study aimed to examine the cause of and effective measures against cluster infections, including the delta AY.1 variant of novel severe acute respiratory syndrome coronavirus 2 (SARS-CoV-2) that occurred in an accommodation facility. Methods: We surveyed the zoning and ventilation systems of the cluster accommodation, examined the staff’s working conditions, conducted an interview, and administered a SARS-CoV-2 test (positive samples were further tested with molecular biological test). Results: Among the 99 employees working at the accommodation, 10 were infected with the delta AY.1 variant. The causes of the cluster infections were close-distance conversations without an unwoven-three-layer mask and contact for approximately five minutes with an unwoven mask under hypoventilated conditions. Conclusions: The Delta AY.1 infection may occur via aerosols and an unwoven mask might not prevent infection in poorly ventilated small spaces. Routine infection detection and responding quickly and appropriately to positive results helps to prevent clusters from spreading.

## 1. Introduction

Since the identification of the novel severe acute respiratory syndrome coronavirus 2 (SARS-CoV-2) in Wuhan in 2019 [1], mutations in the virus have successively appeared [2] and have become a global threat. In Japan, in late April 2021, the proportion of individuals infected with the alpha variant (alpha) of the SARS-CoV-2 B.1.1.7 strain increased, and this strain has nearly replaced the conventional 20B strain [3]. Moreover, the delta variant (delta) of the B.1.617.2 strain has been sporadically detected since May and an increasing trend has been reported [4].

The delta variant is associated with a higher rate of secondary infections than the alpha variant, and individuals who have received only a single dose of the vaccine show an increased susceptibility [5]. Reduced vaccine efficacy against the 2019 coronavirus disease (COVID-19) and increased risks of hospitalisation and reinfection have been reported [6]. Although Australia was initially successful in containing the spread of SARS-CoV-2, the virus eventually spread across the country because of the emergence of the delta variant [7]. In the United Kingdom, as of June 2020, the delta variant has been identified as the predominant SARS-CoV-2 strain [8]. Delta variants with L452R and E484Q mutations have been reportedly identified [9] and the AY.1 strain with the K417N mutation, which is significantly more infectious, has been reported in India and has been referred to as ‘delta plus’ by the Indian Ministry of Health [10]. In Japan, delta AY.1 was first detected in late May; however, as of July 2021, it has not yet become the mainstream SARS-CoV-2 strain [11].

In Japan, individuals who test positive for SARS-CoV-2 can either stay at home or at an accommodation facility, such as a hotel, while undergoing treatment. In Kanagawa Prefecture, seven hotels have served as accommodation facilities for patients with COVID-19 (as of July 2021). In May 2021, a cluster infection with the delta AY.1 variant (in Japan, a cluster has five or more infected individuals) occurred among the staff working in one of these facilities. Identifying the occurrence of delta AY.1 cluster infections is necessary for efficient and effective infection control measures. Thus, we investigated how, in what situations, and through which pathways the cluster occurred. The present study is significant in that it investigates measures to prevent infections and clusters caused by the delta AY.1 variant.

## 2. Materials and Methods

This study was based on a cluster of staff and patients with delta AY.1 in an accommodation facility for patients with COVID-19 operated by Kanagawa Prefecture. The duration of the investigation was based on the day of the onset of COVID-19 in a staff member (day 0), with day 14 as the starting period and day 24 as the ending period (39 days). Since the study targeted individuals who tested positive for SARS-CoV-2, it was considered a case-series.

### 2.1. Working Status of the Staff and Zoning at the Facility

The written work records of the staff were assessed, and a field survey was conducted on the zoning system to prevent patients and staff from coming into contact with each other.

### 2.2. Status of Mutant Virus Detection in Patients Who Remained at the Facility

The number of patients who entered and left the facility and the delta AY.1 detection status were investigated based on the statements of the individual in charge of the facility.

### 2.3. Ventilation Status

The ventilation status of each area was investigated. The location and performance of the air supply/exhaust system and the air conditioning were confirmed from design drawings, and the actual ventilation status was verified using a carbon dioxide concentration meter (CO_2_ Manager, Toa Industry Co., Ltd., Tokyo, Japan). Japanese law requires that the indoor concentration of CO_2_ must be maintained at ≤1000 ppm; hence, this value was used as an indicator of the ventilation status. Eleven CO_2_ concentration meters were installed at a height of 70 cm from the floor in each area (Figure 1), and after measuring the CO_2_ concentrations (baseline), the areas were filled with 2000 ppm of CO_2_ using a CO_2_ gas cylinder. Once the maximum value was reached, the CO_2_ concentration was measured every 2 min until it fell to the baseline concentration.

The CO_2_ concentrations were analysed using one-way analysis of variance by considering the time course in each area. Statistical analyses were performed using the free statistical software EZR version 1.50 (Jichi Medical University, Tochigi, Japan) [12].

### 2.4. Testing and Diagnosis

During the investigation period, any staff members suspected of having COVID-19, such as those with fever or cold, were instructed to undergo polymerase chain reaction (PCR) testing for SARS-CoV-2. For the asymptomatic staff, two PCR tests were performed during this period. PCR testing was performed in accordance with the method published by the National Institute of Infectious Diseases [13].

Mutant viruses were identified by the National Institute of Infectious Diseases or the Regional Institutes of Public Health only when the Ct value from the PCR testing was ≤30 [14]. The remaining samples were subjected to Sanger sequencing using an Applied Biosystems 3500 Genetic Analyzer (Thermo Fisher Scientific K.K., Tokyo, Japan), and whole-genome sequencing was performed using iSeq 100 (Illumina Inc., San Diego, CA, USA) for samples with a high viral load.

### 2.5. Staff Interview

A public health nurse based in Kanagawa Prefecture retrospectively interviewed the staff either face-to-face or via telephone regarding their contacts with PCR-positive individuals. The initial interview was conducted within 1 week of testing positive or negative.

### 2.6. Ethical Considerations

The research protocol for this study was approved as control number 2021 by the Institutional Review Board of the Saiseikai Yokohamashi Tobu Hospital. This research was funded by Health, Labour, and Welfare Science Research Grants (Grant Number: 20CA2022).

## 3. Results

### 3.1. Working Status of the Staff and Zoning at the Accommodation Facility

The lodging facility was a 234-room business hotel. The staff operating the accommodation facilities were classified into four groups: prefectural staff, nurses, housekeepers, and security guards. The total number of staff members working throughout the cluster period was 99. The accommodation facility was zoned into two major areas: the area where patients temporarily stayed (area D) and the ‘no patient access’ area (Figure 1). Furthermore, there was a barrier or wall at the boundary of the two areas, whose upper part did not reach the ceiling, thereby facilitating the passage of air.

### 3.2. Status of Mutant Virus Detection in Patients Staying at the Facility

Table 1: The total number of patients was 1419, and the mean number of patients per day was 51. Not all the patients were tested for delta AY.1, but there were no reports on delta AY.1 or delta detection from these untested patients or from infected individuals associated with them.

### 3.3. Ventilation Status

The layout and ventilation status of the ventilation systems in the areas where each staff member was present, including area D, are shown in Figure 2. The areas for the prefectural staff (area A), nurses (area B), and housekeepers (area C) were connected through corridors. Each ceiling fan (exhaust port) was designed to have a different air volume displacement; however, one ceiling fan in area A was out of order. The number of individuals who could stay in each area were 13, 24, and 14 in areas A, B, and C based on the required displacement (30 m^3^/h per person) in accordance with Japanese law.

CO_2_ concentrations were measured in each area. The outdoor CO_2_ concentration was approximately 430 ppm, and the error range of each measuring instrument was 0–50 ppm. At the time of the cluster’s occurrence, the air conditioners in area A were not being operated, as opposed to those in areas B, C, and D, which were operational use. Therefore, the conditions at the time of measuring the CO_2_ concentration were set identically to those at the time of the cluster infection.

Changes in the CO_2_ concentration at each measurement point are depicted in Figure 3. The decrease in the CO_2_ concentration at the start of the measurement was rapid in area B but showed a gradual and decreasing trend. The ventilation range in area A was within legal requirements, but the ventilation was slightly worse in area A because the exhaust port fan was broken.

### 3.4. Testing and Diagnosis

Thirteen staff members were identified with symptoms suggestive of COVID-19, of which nine were required to visit a medical institution and undergo PCR testing, where they tested positive. The remaining four staff members did not visit a medical institution because the onset was on the day of the group testing. Instead, these individuals underwent PCR testing during group testing (Figure 4A), and three of them tested negative and one tested positive. As the three members also tested negative in a separate session that was conducted 2 days after the initial tests, the infection was ruled out.

Genomic analysis was performed with the samples of the 10 PCR-positive staff members, except for 1 member (who had received one vaccination dose) with a low viral load in the sample. Sanger sequencing analysis confirmed that the virus detected in the remaining nine members possessed K417N and L452R mutations that were characteristic of delta AY.1. Moreover, whole-genome sequencing revealed that the samples from the five members with sufficient viral loads contained viruses with genomes that were molecularly homologous, did not exhibit single nucleotide polymorphisms, and thus had completely identical molecular structures. The glycosylation mutation information for SARS-CoV-2 spike proteins in these five samples and the accession IDs registered in the Global Initiative on Sharing Avian Influenza Data [GISAID] (https://www.gisaid.org/, accessed on 6 July 2021) are shown in Table 2.

### 3.5. Staff Interview

A line listing of the facility staff who tested positive is provided in Table 3, and the related epidemiological curves are depicted in Figure 4B. The first episode occurred in a nurse (Case 1). The only contacts between Case 1 and the prefectural staff (Case 2 or Case 3) were for an approximately 5 min conversation on work-related tasks in area B, during which everybody wore nonwoven masks, and for an approximately 5 min contact during staff meetings (up to 26 individuals) twice daily in area A (all staff wore nonwoven masks).

The contact between Case 1 and the housekeepers (Cases 4 and 6) lasted for approximately 10 min, whereas wearing and removing the personal protective equipment for approximately 5 min during the aforementioned general meeting and during daily communications on work-related tasks.

Other possible infection risks included telephone calls for approximately 30 min per day by the prefectural staff in area C where the housekeepers stayed, masks being moved out of position, and conversations during light meals with individuals in the same occupation and those during routine work while wearing nonwoven masks.

The vaccination status, symptoms at onset, oxygen saturation on admission, and prognosis of the PCR-positive staff are shown in Table 4. Approximately 3 weeks before the onset, Case 4 received one dose of the BNT162b2 (Pfizer-BioNTech) vaccine against COVID-19. The remaining individuals who tested positive were not vaccinated.

### 3.6. Measures Taken after the Identification of Positive Individuals

On day 8, the infection countermeasure team from Kanagawa Prefecture visited the facility to ascertain the situation and to instruct all staff members to wear N95 respirators at all times. N95 respirators are primarily used to prevent infection in the wearers [15]. However, because these respirators have a higher filtration ability and a better fit than the nonwoven masks and have been reported to be effective in preventing the infection [16], a mandate to wear them was issued.

## 4. Discussion

This example situation demonstrates that conversations at short distances with nonwoven masks either off or out of position or crowding in a hypoventilated state may be significant risk factors for infection with SARS-CoV-2. Furthermore, depending on the ventilation status, there is a possibility of infection during short-term (approximately 5 min) contact even within the ventilation standards specified by Japanese law. Although it cannot be concluded that conversations with nonwoven masks, even if worn appropriately, are associated with a high risk of infection, it should not be excluded.

SARS-CoV-2 infection may be transmitted via aerosols [17,18,19]. The risk is exacerbated if the distance is short [20]. Consequently, in this example, the infection was likely to have spread mainly via aerosols. In all the cases, it is inferred that the cause of the cluster infection was the environment in which an exposure to aerosols occurred at short distances, although the exposure times varied. Cluster infections caused by the air circulation of air conditioners in poorly ventilated small spaces [21] and infections between guests in adjacent rooms at a hotel without hallway ventilation are highly consistent with the present study [22]. Furthermore, there were cases in which the infection occurred despite wearing nonwoven masks and eye protection on a daily basis [23]. A recent experiment showed that nonwoven masks only filtered 38.5% of the aerosols when worn normally [24], and there was a possibility that the aerosol entered through the gaps in the nose and cheeks, thus resulting in an infection. From this example, it is assumed that nonwoven masks provide limited protection against SARS-CoV-2 infections; thus, the use of N95 respirators with higher aerosol exposure prevention than nonwoven masks should be considered [25].

Wearing a mask can effectively protect individuals against SARS-CoV-2 infection; however, the complete prevention of infection is difficult [26]. In this example, a maximum of 26 individuals had to coexist in the same area for 5 min twice daily in a space 15.8 ([300 + 110]/26) m^3^/h (area A), where only 13 individuals were allowed as per Japanese law. Despite the short duration of these meetings and the fact that all the staff members wore nonwoven masks, the risk of infection was high. While a case of infection with conventional SARS-CoV-2 has been reported at a ventilation rate of approximately 0.9 L/s (0.324 m^3^/h) without wearing a mask [27], the infection in this example managed to spread even though this establishment had a 50-fold increased ventilation rate. This spread can be justified by the higher infectious nature of the delta AY.1 variant, which contains an additional K417N mutation.

This example was a cluster infection with delta AY.1. The delta variant is associated with a higher rate of secondary infection than the alpha variant [28]; hence, the infection might have spread because of the higher infectious nature of delta AY.1. All four cases were in contact with one another for a very short time while appropriately wearing nonwoven masks, thereby suggesting the effect of delta AY.1. However, other infection pathways could also be adequately accounted for in the case of conventional viruses; thus, the routine implementation of conventional measures is likely to contribute to the prevention of large-scale cluster infections.

SARS-CoV-2 might have been spread in this facility by a staff member without disease-related symptoms. However, none of the staff members who were confirmed to be PCR-positive during the group testing performed from day 6 were asymptomatic during that period, and physical deconditioning was not reported in any of the staff members before the cluster infection. Hence, the possibility that SARS-CoV-2 was introduced by asymptomatic pathogen carriers was eliminated. We could not exclude the possibility of infection from patients who stayed overnight, but the risk of infection was estimated to be low. The reasons were that there had been no cases of delta AY.1 infection in Kanagawa Prefecture at the time this cluster infection occurred. Moreover, it was unlikely that patients admitted at that time had delta AY.1. Regarding the passage of air observed near the ceiling of areas C and D, the possibility that the event originated from a housekeeper staying in area C was unlikely, and the transmission of the infection from the patients to the housekeepers is negative.

SARS-CoV-2 is highly infectious on the day of onset and for approximately one or two days before or after the day of onset [29]. Thus, Case 9 who was working on the day of onset and Case 10 working on the day before the onset might have infected the individuals around them; however, no secondary infections from Cases 9 and 10 were verified. The most significant factor considered to have contributed to the absence of infection was that Cases 9 and 10 wore N95 respirators, which probably prevented the spread of infection. Hence, the risk of infection could be reduced if individuals who come into contact with infected patients and who are at risk of infection wear N95 respirators.

This study has three limitations. First, molecular analyses of the virus detected in all patients before and after the onset of the cluster infection were not performed, and the relationship between this example and the infection transmission route in patients was unknown. Second, the reliability of the time of contact and the detailed contact status of the staff members working during the cluster infection could not be guaranteed. Thus, this cluster is specific to this example and cannot be generalised. Finally, whether Case 4 was infected with delta AY.1 and whether some cases shared the same molecular homology could not be established. However, considering the infection status and the contact history, it is highly likely that all the cases were infected with the same virus. Given the infection status and the absence of other risk factors, the cases were assessed as delta AY.1.

## 5. Conclusions

SARS-CoV-2 has been suggested to be an airborne infection transmitted via aerosols. The delta AY.1 variant may be infectious in enclosed spaces with poor ventilation, even if both infected and exposed individuals wear nonwoven masks. Additionally, the use of N95 respirators may be an effective countermeasure to prevent infection. However, for this protection to be guaranteed in the event of a cluster outbreak, the infection status needs to be evaluated based on rapid and extensive mass testing and N95 respirators need to be worn to reduce the spread of infection in the event of a personnel shortage.

## Figures and Tables

**Figure 1 ijerph-19-09270-f001:**
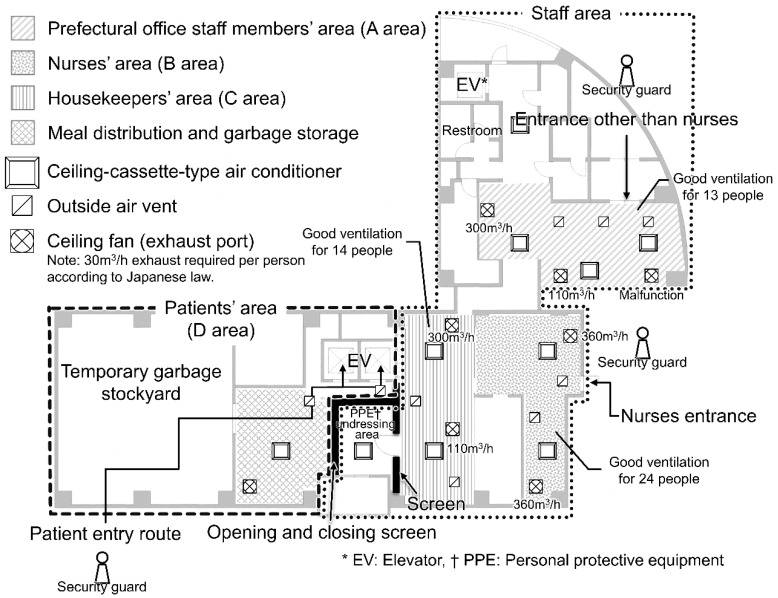
Floor plan and ventilation at the accommodation facility (1st floor).

**Figure 2 ijerph-19-09270-f002:**
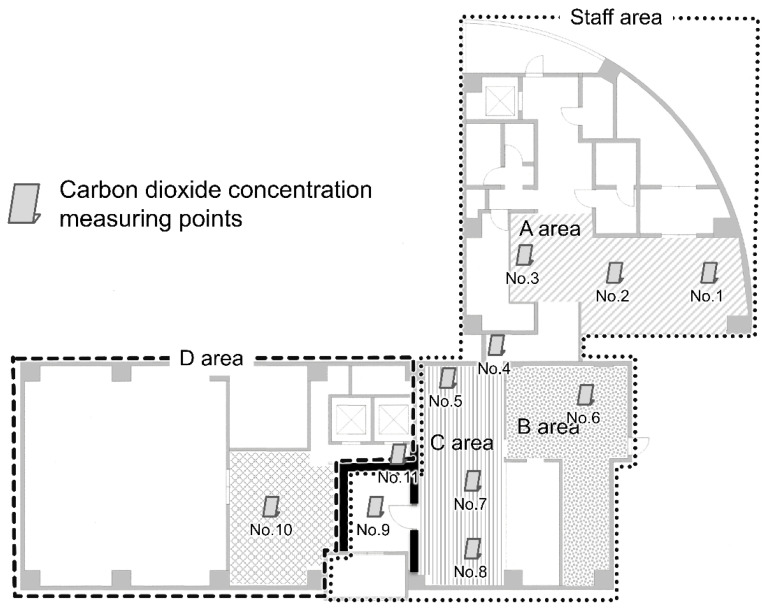
Installation position of the carbon dioxide measuring points.

**Figure 3 ijerph-19-09270-f003:**
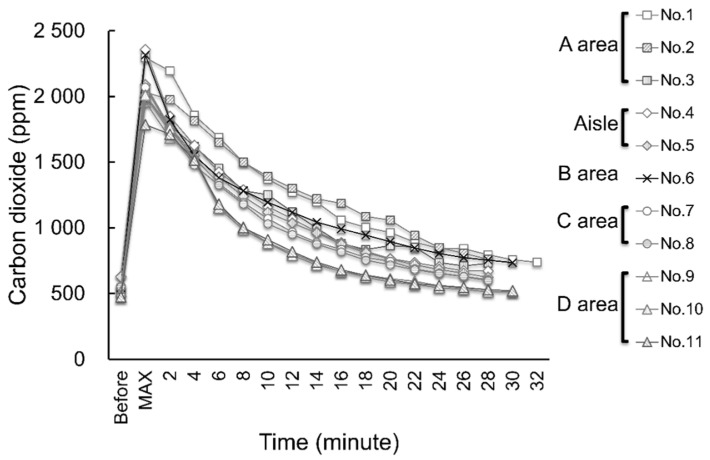
Changes in carbon dioxide concentrations at each measurement point over time. Area A—Prefectural office staff area; area B—Nurses’ area; area C—Housekeepers’ area; area D—Patients’ area. *p* = 0.354 for changes over time in each area (one-way analysis of variance).

**Figure 4 ijerph-19-09270-f004:**
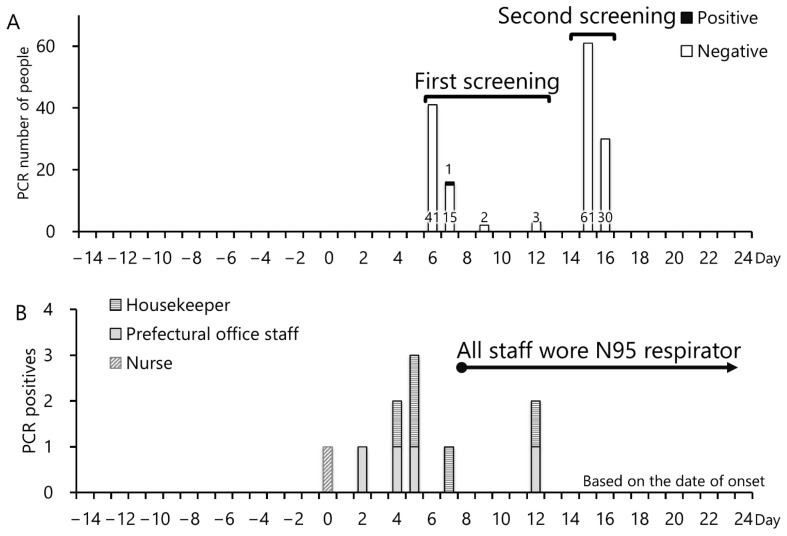
Test timings and onset of disease in polymerase chain reaction-positive individuals. (**A**) COVID-19 epicurve at the accommodation facility. All staff wore N95 respirators from day 8. (**B**) Polymerase chain reaction mass screening test for the accommodation staff. Only one person was positive on the screen test. Nine people were found to be positive outside the screening test.

**Table 1 ijerph-19-09270-t001:** Transition of the number of patients accepted at the accommodation facility during the cluster period.

Case No.	Day-14	Day-13	Day-12	Day-11	Day-10	Day-9	Day-8	Day-7	Day-6	Day-5	Day-4	Day-3	Day-2	Day-1	Day 0	Day 1	Day 2	Day 3	Day 4	Day 5	Day 6	Day 7	Day 8	Day 9	Day 10	Day 11	Day 12
Number of newly accepted patients	1	7	1	1	0	8	10	10	6	5	8	14	15	10	10	7	3	19	13	11	0	0	0	0	0	0	0
Number of discharged patients	12	6	6	5	9	7	7	4	5	5	2	1	3	6	13	9	5	6	10	6	21	12	5	10	10	9	22
Total number of patients accepted	46	47	42	38	29	30	33	39	40	40	46	59	71	75	72	70	68	81	84	89	68	56	51	41	31	22	0

Day 0 was the day of the onset of the disease. The accommodation facility was temporarily closed on Day 12.

**Table 2 ijerph-19-09270-t002:** GISAID accession ID and amino acid mutations in the spike glycoprotein.

Case No.	Accession ID	Spike Glycoprotein												
		19	95	142	156	157	158	258	417	452	478	614	681	950
1	EPI_ISL_3191707	R	I	D	G	del	del	L	N	R	K	G	R	N
2	EPI_ISL_3191716	R	I	D	G	del	del	L	N	R	K	G	R	N
5	EPI_ISL_3191717	R	I	D	G	del	del	L	N	R	K	G	R	N
9	EPI_ISL_3191718	R	I	D	G	del	del	L	N	R	K	G	R	N
10	EPI_ISL_3191719	R	I	D	G	del	del	L	N	R	K	G	R	N

GISAID: Global Initiative on Sharing Avian Influenza Data, del: deletion.

**Table 3 ijerph-19-09270-t003:** Line list of SARS-CoV-2-positive individuals among the staff of the accommodation facility.

Case No.	Occupation	Day-2	Day-1	Day 0	Day 1	Day 2	Day 3	Day 4	Day 5	Day 6	Day 7	Day 8	Day 9	Day 10	Day 11	Day 12	Day 13	Day 14	Day 15	Day 16	Day 17
1	N	NS	NS ^a^	O	SC	P															
2	P	NS			DS	O		SC	P												
3	P	DS	DS	DS				O	SC	P											
4	H	DS			DS			O			SC		P								
5	P		DS		NS			DS	O		SC	P									
6	H	DS	DS	DS	DS	DS	DS	DS	O		SC	P									
7	H			DS		DS	DS	DS	DS O		SC		P								
8	H			DS	DS		DS				O SC	P									
9	P			DS	DS	DS	DS	DS	NS			DS	DS			O SC		P			
10	H	DS	DS	DS	DS	DS		DS	DS	DS	DS	DS	DS		DS	O		SC			P

N—Nurses; PS—Prefectural office staff; H—Housekeeper; DS—Day shift; NS—Night shift; O—Onset (fever, sore throat, etc.); P—PCR-positive; SC—Sample collection for PCR test; Gray—Infectious period considering the incubation period. ^a^ Night shift at another accommodation facility.

**Table 4 ijerph-19-09270-t004:** Vaccination status, symptoms at onset, oxygen saturation on admission, and prognosis of PCR-positive staff.

Case No.	Age	Sex	Vaccination Status	Symptoms at Onset	Oxygen Saturation on Admission	Prognosis
1	43	Female	None	Fever	91%	Good
2	62	Male	None	Sore throat	92%	Good
3	63	Male	None	Fever	Not hospitalised	Good
				Cough		
				Headache		
4	51	Female	Received one dose	Cough	Not hospitalised	Good
5	52	Male	None	Fever	93%	Good
6	43	Male	None	Fever	Not hospitalised	Good
				Malaise		
7	21	Male	None	Sore throat	Not hospitalised	Good
8	20	Male	None	Sore throat	Not hospitalised	Good
9	22	Male	None	Asymptomatic	Not hospitalised	Good
10	36	Female	None	Fever	91%	Good

## Data Availability

Data is contained within the article.

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
