# Peer review of "SARS-CoV-2 Delta AY.1 Variant Cluster in an Accommodation Facility for COVID-19: Cluster Report"

_ijerph, 2022, doi:10.3390/ijerph19159270_

Round 1

Reviewer 1 Report

The paper presents a report on a cluster infection involving the delta AY.1 23 variant of the novel severe acute respiratory syndrome coronavirus 2 at an accommodation facility in Japan.

1. Abstract needs to be formatted properly: typos and formatting errors can be seen. 

2. Result presentation in abstract is very generalised. Should specify correct numbers

3. Ethical considerations should be mentioned initially in methods part

4. Discussion should include similar studies reported from other hong kong or any other specific cluster info and comparison.

5. The implications of this work need to be further elaborated in discussion.

Author Response

The paper presents a report on a cluster infection involving the delta AY.1 23 variant of the novel severe acute respiratory syndrome coronavirus 2 at an accommodation facility in Japan.
Thank you for your constructive and meaningful opinions. We reviewed the comments and made the following corrections. We would be grateful if you could review our corrections.

  1. Abstract needs to be formatted properly: typos and formatting errors can be seen. 
    We corrected typos and added missing letters.
  2. Result presentation in abstract is very generalised. Should specify correct numbers
    We added contact time to the Result section of the Abstract for clarity.
  3. Ethical considerations should be mentioned initially in methods part
    We moved “ethical considerations” to the start of the Methods.
  4. Discussion should include similar studies reported from other hong kong or any other specific cluster info and comparison.
    We discussed the results of comparisons with other cluster infections reported when the delta variant was dominant (Lines 240–242).
  5. The implications of this work need to be further elaborated in discussion.
    Thank you for your comment. The objective of this study was to demonstrate that SARS-CoV-2 infections might spread even with the use of masks if ventilation is poor. We wanted to discuss more about the measures related to that point. However, there is a word limit, and we had to discuss how the current cluster infection was caused by the delta AY.1 variant. Thus, we had to make the difficult decision to reduce the word count on the measures. However, we added a sentence on measures (Lines 241–243) within the word count limit. If you could confirm this change, we would be very grateful.

Reviewer 2 Report

Authors presented interesting topic on a new variant of COVID AY.1. I have a few concerns before it gets accepted

The methods section in the abstract is not clear and presented in a more generalized

Enough literature was presented in the introduction. But the rationale and importance of presenting work are not clear to readers

Patient and staff demographic information missing

I recommend the results section can be rewritten with more precision. Section 3.1 can be moved to the methods section

How about outbreak reproduction rate R0?

Author Response

Authors presented interesting topic on a new variant of COVID AY.1. I have a few concerns before it gets accepted

  1. The methods section in the abstract is not clear and presented in a more generalized
    We rewrote the Methods and Results of the Abstract.
  2. Enough literature was presented in the introduction. But the rationale and importance of presenting work are not clear to readers
    We discussed the rationale and importance in lines 59–62 of the introduction.
  3. Patient and staff demographic information missing
    You are correct. We wanted to discuss more about demographics, but the present objective was to demonstrate that wearing a mask does not prevent SARS-CoV-2 infection if the environment is poorly ventilated. We were concerned that, given the word count limit, discussing the backgrounds and demographics of patients and staff would dilute the focus. These are the reasons why we limited our discussion of demographics.
  4. I recommend the results section can be rewritten with more precision. Section 3.1 can be moved to the methods section
    We moved Section 3.1 to Methods.
  5. How about outbreak reproduction rate R0?
    We thought the R0 of this cluster infection was quite interesting; however, the details of contact time and the reliability of contact situation for each infected employee were not guaranteed. Thus, we avoided calculating the R0 to avoid any bias.

Round 2

Reviewer 2 Report

authors successfully addressed the comments. I recommend to accept this work after minor language corrections